# MOFs-Graphene Composites Synthesis and Application for Electrochemical Supercapacitor: A Review

**DOI:** 10.3390/polym14030511

**Published:** 2022-01-27

**Authors:** Surendra K. Shinde, Dae-Young Kim, Manu Kumar, Govindhasamy Murugadoss, Sivalingam Ramesh, Asiya M. Tamboli, Hemraj M. Yadav

**Affiliations:** 1Department of Biological and Environmental Science, College of Life Science and Biotechnology, Dongguk University-Seoul, 32 Dongguk-ro, Ilsandong-gu, Siksa-dong, Goyang-si 10326, Gyeonggi-do, Korea; shindesurendra9@gmail.com (S.K.S.); sbpkim@dongguk.edu (D.-Y.K.); 2Department of Life Science, Dongguk University-Seoul, 32 Dongguk-ro, Ilsandong-gu, Goyang-si 10326, Gyeonggi-do, Korea; manukumar007@gmail.com; 3Centre for Nanoscience and Nanotechnology, Sathyabama Institute of Science and Technology, Chennai 600119, Tamil Nadu, India; murugadoss_g@yahoo.com; 4Department of Mechanical, Robotics and Energy Engineering, Dongguk University, Seoul 04620, Korea; sivaramesh_74@yahoo.co.in; 5School of Chemical Engineering, Yeungnam University, 280 Daehak-ro, Gyeongsan 38541, Korea; 6School of Nanoscience and Biotechnology, Shivaji University, Kolhapur 416004, Maharashtra, India

**Keywords:** coordination polymers MOFs, graphene, supercapacitors, chemical method, nanomaterials, porous nanostructures

## Abstract

Today’s world requires high-performance energy storage devices such as hybrid supercapacitors (HSc), which play an important role in the modern electronic market because supercapacitors (Sc) show better electrical properties for electronics devices. In the last few years, the scientific community has focused on the coupling of Sc and battery-type materials to improve energy and power density. Recently, various hybrid electrode materials have been reported in the literature; out of these, coordination polymers such as metal-organic frameworks (MOFs) are highly porous, stable, and widely explored for various applications. The poor conductivity of classical MOFs restricts their applications. The composite of MOFs with highly porous graphene (G), graphene oxide (GO), or reduced graphene oxide (rGO) nanomaterials is a promising strategy in the field of electrochemical applications. In this review, we have discussed the strategy, device structure, and function of the MOFs/G, MOFs/GO, and MOFs/rGO nanocomposites on Sc. The structural, morphological, and electrochemical performance of coordination polymers composites towards Sc application has been discussed. The reported results indicate the considerable improvement in the structural, surface morphological, and electrochemical performance of the Sc due to their positive synergistic effect. Finally, we focused on the recent development in preparation methods optimization, and the opportunities for MOFs/G based nanomaterials as electrode materials for energy storage applications have been discussed in detail.

## 1. Introduction

Restrictions in renewable energy resources and the rising air pollution have catalyzed the request for green and clean sustainable energy. To solve this problem, researchers from around the globe have been actively working on the development of a novel composite electrode material for the efficient environmental remediation, conversion, and storage of electrical energy in the form of electrochemical energy [1]. In the electric market, the main two systems are available batteries and supercapacitors (Scs). Among these electrochemical energy storage devices, batteries store charge via the oxidation/reduction of electrode-active materials and/or intercalation/de-intercalation of the ions into/out of the electrode materials. Batteries can provide a higher energy density; however, the rate capability of the batteries is kinetically limited by the slow redox processes and the slow ion transfer through the electrode-active materials; thus, the power density of batteries is usually poor. The energy-storage mechanisms of the Sc involve two-step processes: the first is related to the electrochemical double-layer capacitance (EDLC) based on the physical adsorption of charges at the solid-liquid interface and the second is associated with the pseudocapacitance based on the redox reaction on the surface of electrode materials [2,3,4,5,6].

Among these various energy-storage systems, Sc have significantly grown consideration in recent few decades due to their smart applications such as fast charge/discharge of the ions, higher energy, power density, better cycling stability rates, and eco-friendly environmental atmospheric. Recently, Sc have been widely used in several portable electronic applications in the worldwide market e.g., e-bike, metal forming, racing starter, camera flash, electric vehicle charging sub-stations, thermonuclear, conventional bus truckers, in the military vehicle ICE starting, hybrid vehicles regeneration, wind turbine output load cranes, UPS, forklifts, emergency doors on an Airbus A380, and back-up power [7]. To fill this gap of the two energy storage devices such as capacitors and batteries, researchers designed new and advanced electrode active materials to improve the diffusion kinetics and provided a higher active surface area [8].

Various types of micro/mesoporous nanomaterials are reported in the previous studies; out of these, MOFs presented superior to the other nanomaterials because the MOFs show better structural, surface morphology, and electrical properties. In addition, it shows superior cycling and thermal stability, allowing them to control their pore size and nanostructure/shape. All these physicochemical properties have attracted a lot of attention over the past two decades for a wide range of potential applications in electrochemical Sc. MOFs are attractive candidates to perfectly meet the needs of next-generation energy storage technologies [9]. MOF nanomaterials provide a higher specific surface area (SSA) of around 10,000 m^2^ g^−1^, which is much higher than other nanomaterials. MOF constructions and applications were prepared and emphasized in the 1990s by various researchers owing to their excellent electrochemical properties [10]. 

Recently, various previously studied porous nanomaterials, e.g., carbon-based nanomaterials such as carbon nanotubes (CNTs) and G, have received great attention due to their high SSA and electrical conductivity. G-based composites with MOFs are particularly promising due to the possibility to achieve synergic effects between the porous MOFs and higher SSA of the G [11]. G work as an integral component of the framework to reduce the over potential of the working electrode. On the other hand, the faster transfer of ions of G nanosheets in the MOFs framework is very important to make it a suitable super way for electron transfer [11]. According to the structural versatility of MOFs, it is crucial to study the role between structural properties and electrochemical characteristics of various MOFs, including porous coordination networks, two-dimensional (2D), entangled, polyoxometalate, heterometallic, and novel emerging MOFs [12]. Because of these interesting features, MOF/G-based materials have been considered promising for various applications and can be prepared by various methods such as solvothermal, co-precipitation, ultrasonication, physical mixing, reflux, in situ, and post-synthesis. 

A few reviews have discussed MOF/G composites systems for electrochemical Sc applications. For instance, Serre et al. [10] reviewed different synthetic methods and challenges towards the characterization of MOF/G systems. Similarly, Pang et al. focused on the preparation and application of these systems [13]. In contrast, Lai et al. reported electrocatalysis and photocatalysis applications of these composites [14]. Liu et al. [15] reported advances used for the optimization of electrochemical performance of MOF-derived metal oxides in Sc. In recent years, several reviews are available concerning electrochemical Sc of MOFs [12,16], but to the best of our knowledge, few of them focus on MOF/G based electrochemical Sc and their synthesis methods. 

In the present review, we have focused on reviewing the current development of different chemical synthesis methodologies for MOF, metal-based MOFs, MOF/G, MOF/GO, and MOF/rGO composites nanomaterials for electrochemical Sc applications. The synthesis method and the effect of different experimental parameters such as the percentage of GO, nanostructure, design, and assembly of asymmetric/symmetric of SCs have also been covered along with their structural, morphological, and electrochemical studies. We also focused on the electrochemical characteristics of these composites based on published reports. 

## 2. Electrochemical Capacitor

Carbonaceous materials are classical while MOFs are considered state of the art materials; therefore, the amalgamation of these materials is found to be fascinating. MOFs based Sc can be distinguished as: (a) pure MOFs materials used for charge storage on their internal surfaces through a non-faradaic mechanism or the redox reaction of the metal center is utilized to store energy; (b) MOFs are pyrolyzed to obtain porous carbon nanomaterial and their capacitance can be booted with the aid of conventional EDLCs materials, and (c) MOFs are converted into metal/metal-oxides, which store energy in the form of the faradic mechanism, which is responsible for the faster charge/ions transfer using the electrochemical reaction [17,18]. 

### 2.1. Electrochemical Double-Layer Capacitor

The electrochemically inactive materials with very high specific surfaces are mostly used in EDLC to develop the performance of the Sc. Recently, G and MOFs have more attractive nanomaterials due to their electrical properties, such as a higher active surface area and the porous surface morphology of the electrode. The mechanism of charge storage includes the accretion of charge on the surface of the electrode–electrolyte interface. The advantage of EDLC is ultra-high power density and an excellent cycling stability rate. The specific capacitance (Cs) and energy density in EDLC are mainly affected by the effective SSA of the working electrode.

### 2.2. Pseudocapacitor

The charge storage mechanism of pseudocapacitors includes faster and reversible faradaic reactions on the surface of the electrode–electrolyte interface. The main advantage of the pseudocapacitor is providing more Cs and higher energy densities of the active electrode material. Despite this outstanding electrical property, pseudocapacitors demonstrate poor electrical conductivity of the electrode material. Metal oxides and conducting polymers show faradic behavior and possess redox reactions. These reaction mechanisms are parallel to battery-type electrode materials and have less cycling stability as well as low power density. 

### 2.3. Hybrid Capacitor

The amalgamation of EDLC and pseudocapacitor materials is known as a hybrid capacitor which offers excellent cycling stability, power density, and higher Cs values. All battery-type electrodes included in the HSc, e.g., all transition metal oxide/hydroxide, especially Ni-based electrodes and porous carbon, provide excellent potential windows to deliver higher energy density than other capacitors and EDLCs. In addition, the HSc supplies a broader potential window and higher capacitance [19,20]. 

The previous literature indicates that the reported values of the capacitance of the MOFs are lower than that of transition metal oxides because of the lower electrical conductivity and compact-like surface morphology, which is not suitable for the faster transformation of the ions in the electrolyte. To solve these problems, researchers are working on the doped carbon, metal oxides, or composite of the materials such as metal oxide carbon, which are helpful for reducing the particle size and porous-like nanostructure [17].

## 3. Synthesis Methods for MOF/G-Based Materials

All researchers know that the electrical conductivity and higher SSA of active electrodes are important parameters to improve the electrochemical performance of the as-prepared electrodes. The nanomaterials are amalgamated on the conducting electrode material such as rGO, G, and carbon fiber paper (CFP) to increase the electrical conductivity of the electrodes. Therefore, MOFs are amalgamated with conductive materials as additives, which are more capable alternatives for devolvement in the prepared composite materials for high electrochemical performance [17]. Generally, the preparation of MOFs/GO composite is the former process of the preparation of MOFs-derived nanomaterials. Several approaches have been developed to prepare MOFs/GO composite, such as solvothermal, co-precipitation, ultrasonication, mixing, reflux, in situ, and post-synthesis methods.

### 3.1. Solvothermal Synthesis Method

The solvothermal method has been widely used for the preparation of MOFs/GO nanocomposites electrode materials due to its greater advantages compared to the other chemical methods, such as easily controlling the size and shape of the nanostructures; these types of surface morphology provide higher porous nanostructures. The high porosity and greater SSA of the nanomaterials make them more capable of Sc application. However, this method has a few drawbacks, such as the requirement of high temperatures and more time to complete the reaction/growth of the nanostructure. 

Jiao and co-workers [17] report on the nickel-based pillared MOFs with rGO (ADC-rGO) composite synthesis through a facile solvothermal reaction method (Figure 1). Initially, GO was exfoliated in dimethylformamide (DMF) solution and then sonicated with a stoichiometric amount of Ni(NO_3_)_2_·6H_2_O, 9,10-anthracenedicarboxylic acid (H_2_ADC), and 1,4-diazabicyclo[2.2.2]octane (DABCO) solvents. After sonication, the prepared product was placed in the oven at 120 °C for 48 h for preheating to complete the development of the nanostructures on the MOF electrode material. The surface morphology of ADC-rGO and ADC-OH/rGO show the structural devolvement in the nanostructures from solid micro rods into hollow and porous micro rods was observed after KOH treatment. After that, they prepared Ni^2+^ metal ions at a strong alkaline solution, dissociated them with the organic ligands (ADC and DABCO) and also subordinated them with OH^−^ ions, leading to the breakdown of the metal–ligand coordination bonds. As the reaction started, the OH^−^ ions continued to diffuse and react with the inner core of ADC-rGO; meanwhile, newly designed Ni(OH)_2_/ADC-rGO was constantly deposited on the as-preformed Ni(OH)_2_ surface resulting in the formation of hollow micro rods. Finally, ADC-rGO was carbonized to obtain ADC-C/rGO under the N_2_ atmosphere. 

In the previous literature, one more report is available on the solvothermal method for the preparation of the MOF. Azadfalah et al. [11] prepared G and Cu-MOF (G@Cu-MOFs) nanocomposites of the electrode using the one-step and facile solvothermal method. Here, they dissolved copper nitrate in double-distilled water (DDW) and benzene-1,3,5-tricarboxylic acid as a linker (BTC) in ethanol using ultrasonication, and transferred the mixture to the autoclave, along with G, and finally heated it at 120 °C for 24 h. They interestingly noted that when a G source is added during the synthesis of Cu-MOFs, the nanostructure becomes more porous, and hierarchical nanoflower-like surface morphology is grown on the surface of the sample. The more porous nature helps to better the migration of ions and electrons. This nanostructure shows that G causes the Cu-MOFs nanostructure to transform from the original octahedral to hierarchical flower-like nanostructures. A similar phenomenon was observed by Yadav et al. [21] after the incorporation of cellulose nanofibers (CNF) into MOFs. CNF and hybrid zeolite imidazole framework (HZ) was developed from biomaterial and porous carbon-based material, respectively. They prepared metallic cobalt-coated N-doped nanoporous carbon (NPC) wrapped with CNTs nanocomposites which were synthesized using a chemical method for Sc application and changing the ratio of CNF in the composite. Finally, they concluded that CNF–HZNPC composite electrodes are better than other electrodes. They reported that the Cs is 146 F/g at a current density of 1 A/g with cycling stability of up to 90% at 10 A/g over 2000 cycles.

Tan et al. [22] reported a synthetic strategy of polyoxometalate (POMOFs) based MOFs on C-POMOFs/rGO nanosheets by a hydrothermal method. They reported the effect of the calcination temperature on the electrochemical performance of the prepared final C-POMOFs/rGO composite. GO nanosheet-like nanostructures were deposited on the conducting substrate and electrons transportation scaffold, the Co^2+^ and PMO_10_V_2_ were slowly adsorbed on the surface of the GO nanosheets and successively coordinated with 2-methylimidazole. Subsequently, after calcination temperature, the surface morphology of the C-POMOFs shows the nanosheet-like nanostructures grown on the surface of the rGO nanosheet. GO nanosheets demonstrate a highly conductive way to increase charge/mass transport during the charging/discharging processes and enhance faradaic processes on the interface of the electrode/electrolyte. Kim et al. [23] prepared Ni-MOFs/rGO composite by a hydrothermal method for electrochemical performance. Initially, GO nanomaterials were synthesized using the modified Hummers method for the base material of the electrode. After successfully preparing the GO nanomaterial, it was directly mixed with nickel nitrate and trimesic acid (H_3_BTC) to produce Ni-MOFs/GO composite electrode material via a hydrothermal method. The transformation of GO to rGO was achieved by heat treatment. Finally, they obtained Ni-MOFs/rGO composites annealed at 250 °C for improved electrochemical performance. The resultant report found that the crystalline Ni-MOFs were uniformly distributed on the exterior of rGO. This rGO can also be used as a substrate and an active material to promote the charge supply of Ni-MOFs nanomaterial. In addition, Ni-MOF was uniformly distributed in the rGO nanosheets to separate neighboring rGO nanosheets as spacers.

The solvothermal route is advantageous compared to other routes and widely utilized to prepare different nanostructures. This route provides a strong linkage between two materials which ultimately enhances electrochemical properties and stability. 

### 3.2. Co-Precipitation Synthesis Method

The co-precipitation method has some merits over the other chemical method and physical methods, such as the ease of preparation, controllability for experimental parameters, high purity, and ability to produce a product in larger quantities. Moreover, materials with a smaller crystallite size and porous surface morphology can be prepared using the co-precipitation method. However, this method requires high-quality solvents, water for washing, and more time for drying the product. Beka et al. [24,25] prepared 2D nanosheet-like nanostructures of the NiCo-MOF electrode material grown on the surfaces of the rGO deposited a room temperature using a precipitation method for higher electrical Sc performance. The highly porous rGO nanosheets are useful to generate a faster pathway for ionic and electronic conduction, while NiCo-MOF/GO porous nanosheets have higher active surface area exposed sites that can act as superb pseudocapacitive electrode material. Sulfur nanoparticles (NPs) are fully and uniformly decorated on the rGO nanosheets by self-assembly deposition using the hydrothermal method. After that, decorated sulfur NPs coated 3D rGO nanosheets are heated using annealing temperature to form 3D porous rGO interconnected networks. NiCo-MOF/rGO hybrid electrode material is prepared using a precipitation method at room temperature. The resultant Cs values show the better electrochemical performance of the NiCo-MOF/rGO electrode materials.

In addition, in this report, the authors described the growth mechanism of the NiCo-MOF/rGO hybrid electrode as follows. During the preparation of the porous rGO nanosheets, in the first step, the HCl was used to produce free sulfur to functionalize the rGO nanosheet nanomaterials. Then, the MOF precursors and rGO are uniformly dispersed in the solution under a constant string process to the functionalized rGO with MOF nanomaterials. The negative charge on the nanostructure sites of rGO nanosheets is very useful for electrostatic attraction between the positive charge of the metal ions of the electrode materials. They are attracted to these surfaces of the nanostructures and start nucleation on the MOFs as a center of the electrostatic growth of the electrode materials. They indicate the density of negatively charged ions depending on the rGO concentration ratio, and it is useful to change the nucleation rate of MOF nanomaterials. In addition, they mentioned that, at a lower concentration of rGO, the composite material shows some limitations such as less negative charge ions at attraction sites and MOFs formed at the nucleation center. Most of the MOF precursors are used during the growth formation, and large nanosheet-like surface morphology is observed. Furthermore, they reported a higher concentration of rGO with MOFs composite. The reported results show that almost all rGO nanosheets are covered by the MOF precursors as a nucleation center, and the remaining amount of MOF precursors active in the growth process are limited. As a result, the size of the nanosheets are smaller when compared with the high concentration of MOFs and rGO ratio in the reported samples. Thus, the balance between nucleation and growth of MOF on the rGO nanosheet supports the preparation of ultrathin 2D nanosheets of MOF on rGO nanosheets. Finally, they concluded that this approach is a capable way to produce and control the size and shape of the nanostructure of the NiCo-MOFs/GO nanosheet of the electrode 

Banerjee et al. [26] synthesized the MOFs/rGO composites material with various weight percentages of rGO (5, 10, 30, and 50%) by dispersing it in a mixture of dimethylformamide (DMF) and chlorobenzene solution (Figure 2). Nickel nitrate, zinc nitrate, and BDC were dissolved in the rGO/DMF/chlorobenzene dispersion solution to prepare Ni, Zn-based MOF/rGO nanomaterials. Further, trimethylamine (TEA) and a few drops of hydrogen peroxide (H_2_O_2_) were added dropwise, and the mixture was agitated for 45 min. After completion of the reaction, the resultant black precipitate was collected by vacuum filtration and washed with DMF solution. The black filtrate product was kept in chloroform to convert the DMF solution for 24 h. 

### 3.3. Ultrasonication Synthesis Method

The ultrasonication method is considered a green, eco-friendly, energy-efficient, and direct synthesis method for the preparation of the MOF/G-based composites material for the electrochemical application [27]. However, the large-scale synthesis approach by ultrasonication method is not common. Hosseinian et al. [28] demonstrated a hybrid composite between zeolitic imidazolate framework (ZIF) and rGO nanosheets by the ultrasonication method at room temperature. ZIF-67 is developed by the linkage of 2-methylimidazolate anions and cobalt cations. The authors used a simple technique to facilitate the dispersion of the GO nanosheet in an ethanol solution and DI water combination at an equal ratio under the constant sonication treatments. After the uniform and continuous dispersion of GO nanosheets in DI water, they added ZIF-67 particles directly and again sonicated at room temperature. Similarly, in another report, the author used as-prepared ZIF-67 nanocrystals to deposit on the surface of the rGO nanosheets using a simple and cost-effective chemically ultrasonic method at room temperature. The Sc properties of the as-prepared ZIF-67 and rGO/ZIF-67 were tested by cyclic voltammetry (CV), galvanostatic charge-discharge (GCD), and electrochemical impedance spectroscopy (EIS) measurements in an aqueous electrolyte. The prepared ZIF-67 and nanocomposites between ZIF-67 and rGO electrodes show Cs values of 103 and 210 F/g at a constant current density of 1 A/g, which indicates the rGO/ZIF-67 electrode provided higher Cs than the pure ZIF-67 electrode at a constant current density with outstanding cycling stability up to 80%. Srimuk et al. [29] prepared different weight percentage ratios of rGO to HKUST-1 MOFs using ultrasonication and the solvothermal method. They observed a pure MOF microporous structure-like surface, but these nanostructures offer lower electrical conductivity. To improve the electrical conductivity of as-prepared pure MOF, the author designed simple strategies to compose it with a different weight percentage of the rGO. The composite demonstrated a very high surface area of 1241 m^2^/g. Finally, they concluded that the 10 wt.% rGO/MOF shows the higher Cs values. The as-prepared 10 wt.% rGO/MOF electrode shows the Cs of the 385 F/g, at current density 1 A/g. For practical application, they constructed an asymmetric solid-type Sc device of the optimized rGO/MOF electrode offering a specific power and energy of 3100 W/kg and 42 Wh/kg, respectively.

### 3.4. Mixing Synthesis Method

The mixed method is very useful for the preparation of the nanomaterials for the supercapacitor application. This method does not require sophisticated equipment and is operated at low temperatures and pressures. However, the agglomerated nanostructures are obtained using this method which is not useful for the electrochemical application. Furthermore, the bonding between the two materials is very weak. Xu et al. [30] reported on the large-scale preparation of various 3D GO/MOF nanocomposites using a mixing method. In this report, the author used as-synthesized MOF and GO nanosheets for the preparation of the composite in three steps. In the first step, MOF and GO are stirred continuously to develop three-dimensional porous nanostructures of hydrogels. In the second step, the prepared hydrogel of the GO/MOF composite was directly converted into an aerogel-like nanomaterial using freeze-drying treatment. Finally, the prepared nanomaterials were heated under N_2_ and air atmospheres, respectively, to convert a pure phase of rGO/MOF formation of the resultant composite in step 3. Note that the two-step annealing process can prevent the prepared rGO/MOF-derived composite aerogels from collapsing and helps to retain the 3D porous structure. Different composite aerogels prepared from GO with Fe-MOF, Ni-MOF, ZIF-8 (Zn cations bridged with 2-methylimidazolate anions), MOF-5, Sn-MOF, Co-MOF, and even as Fe-MOF/Ni-MOF mixture were achieved. The electrochemical results show that the rGO/Fe_2_O_3_ nanocomposite electrode provided a higher Cs of 869.2 F/g than the rGO (360.8 F/g) and Fe_2_O_3_ (258.5 F/g) at a constant current density of 1 A/g, respectively. 

### 3.5. Reflux Synthesis Method

The reflux method has some merits such as ease of preparation, it is eco-friendly, the morphology of the surface is easily controlled by simple experimental parameters, and it is less expensive. The porous surface of nanomaterial improves the electrochemical performance because these types of surfaces provide easy paths for the ions moving from the electrolyte/electrode interface. Moreover, the reflux method is based on the solution-based chemical method, so a solid-state reaction is not possible. Wang et al. [31] refluxed GO material, zinc nitrate, and 1,4-benzene dicarboxylate (H_2_BDC) in DMF solution. The final product was treated with DMF and chloroform. MOF-5 and MOFs-5/GO were heated at 270 °C in the air, followed by N_2_ at 800 °C, and washed with HCl; much time was taken to cleanse all impurities. The framework-like nanostructure of MOFs-5 shows 3D interconnectivity with [Zn4O]^6+^ clusters arranged at eight corners of a cube and linear 1,4-benzene dicarboxylate (BDC) linkers placed on the edges of the cube. When GO was incorporated into MOF-5, epoxy and carboxylic groups on the GO reacted with [Zn_4_O]^6+^ clusters by oxygen replacement and thin GO nanosheets acted as dividers located in between the thin platelets of MOF-5. This led to the formation of an ordered and sandwich-like structure of MOF-5/GO nanosheets (Figure 3). Moreover, the interaction between GO and MOF-5 crystals was greatly enhanced by hydrogen bonding between hydroxyl groups on GO and oxygen atoms in the ZnO_4_ tetrahedra. The final composite showed a regular cube-like nano shape with uniform assembling on the GO sheets and an average size of about 10–20 µm. 

Rahmanifar et al. [7] synthesized Co-MOF, Ni-MOF, and Ni/Co-MOF/rGO nanocomposite rGO samples for high-performance Sc applications. They prepared Co-MOF/rGO and Ni/Co-MOF/rGO composite with various percentages (wt.%) of rGO (20, 40, 50, 60, and 80%) by reflux method at 140 °C for 72 h. The composite of Ni-MOF and Co-MOF/rGO with wt.% ratios of 25:25:50 was activated, and electrochemical measurements were performed. The final composite of the Ni/Co-MOF/rGO presents Cs of 860 F/g at a constant current density of 1.0 A/g, which is higher than the other prepared Co-MOF and Ni-MOF electrode. These improvements in the electrochemical performance may be due to the highly porous nanosheets of rGO. They prepared the asymmetric Sc with activated carbon as a negative electrode material and Ni/Co-MOF/rGO electrode as positive electrode materials. They prepared devices offering specific energy of 72.8 Wh/kg at 850 W/kg with a better cycle stability of 91.6% after 6000 at a constant 1 A/g charge-discharge.

### 3.6. In-Situ Synthesis Method

The in-situ method is preferred to prepare MOF/G-based composites as the linkage between the two materials is strong enough. The nanomaterial prepared using this method provided more advantages for electrochemical application, such as higher electrical conductivity, low resistance, better stability, and higher electrical properties.

Though the in-situ method has been mostly used to develop MOFs/GO composites by many researchers, it has few limitations, such as aggregation of G nanosheets, lower SSA, and unreacted reactants. 

The in-situ approach has been followed by several researchers to prepare MOFs/G composites for various electrochemical applications. Xin et al. [32] prepared G and N-doped porous carbon (HPNCs/rGO-800) nanomaterials for simple carbonization with ZIF-8 by controlling the preparation temperature (Figure 4). They concluded that the aggregated ZIF-8 resulted in poor surface activity while the incorporation of GO helps to reduce aggregation and increase electrical conductivity. The CV, CGD, and EIS results are supported to this conclusion. The HPNCs/rGO-800 electrode displays outstanding supercapacitive properties with a higher Cs of 245 F/g at a constant current density of 1 A/g. Furthermore, they manufactured an asymmetric Sc of the as-prepared HPNCs/rGO-800 electrodes, providing better energy density (18.0 Wh/kg) and a power density (475 W/kg) with good potential voltage (0–1.8 V).

He et al. [19] reported an in-situ polymerization method for the preparation of the hybrid composite nanomaterial CuOx/mC/PANI/rGO. In this report, the author prepared and reported the copper-based MOFs with a combination of the GO and PANI for the electrochemical application. The final product of the MOFs is derived with copper, copper oxide/mesoporous carbon (CuOx/mC) embedded with PANI, and rGO. At room temperature, the CuOx/mC was polymerized using aniline in HCl and (NH4)_2_S_2_O_8_. Then, rGO was introduced into CuOx/mC@PANI to obtain the final hybrid porous nanostructures (Figure 5). The surface morphology shows that the homogeneous and highly porous coating of the PANI with rGO nanosheets was decorated on the surface of the CuOx/mC nanomaterials. Here, PANI and rGO were used to increase the conductivity of the composite electrode. They show the prepared ternary CuOx/mC/PANI/rGO composite reported Cs of 534 F/g with superb stability. The existence of the conductive system improves the diffusion of ions and fast redox reaction on the surface of the electrode, which eventually boosts supercapacitive performance [19].

### 3.7. Post-Synthesis Method

Post-synthesis is an easy approach to prepare MOF/G-based composites for the electrical properties study. The drawback of this method is the poor attachment of the two materials in the composite, so this technique is not widely used to prepare MOF/G-based composites. Furthermore, this leads to the migration of the material from the pores to the surface, in addition to the reduction of surface area, pore volume, and pore diameter. 

Cao et al. [33] prepared rGO/MoO_3_ composite by mixing GO with Mo-MOFs followed by a two-step annealing treatment under Ar and atmospheric air conditions. As-prepared GO was thermally converted to rGO during annealing in air conditions. After adding rGO nanomaterial, the prepared product provided a higher surface area of the rGO/MoO_3_ composite than the pure MOFs/GO and MoO_3_ nanomaterials. The Cs result of rGO/MoO_3_/MOFs of the 617 F/g at a current density of 1 A/g. Finally, they determined that the as-prepared rGO/MoO_3_ composite shows good cycling stability in Sc application, indicating this nanomaterial is very useful for practical level applications in flexible electronics.

Bai et al. [34] refluxed as-prepared ZIF-67/G composite with nickel precursor to obtain Ni-Co LDH, which was removed by acid treatment. The mass ratios of Ni-Co LDH with 8, 15, and 25 mg of G were about 81, 59, and 25%, respectively (Figure 6). Figure 6B shows the hierarchical hollow nanocages of Ni-Co LDH on the G nanosheet with different magnifications (a–d). The hierarchical sandwich like morphology can provide large active surface for reaction with electrolyte. The mass ratios of G (15 mg) with Ni-Co LDH provided a higher Cs of 1265 F/g and superior cycling (92.9%) after 2000 cycles than the other G percentages. They reported the Ni-Co LDH/G composite material combination of battery-type and Sc electrochemical behavior. In addition, the prepared asymmetric Sc device provides more Cs of 170.9 F/g with excellent energy density and power output. 

## 4. Electrochemical Properties of MOF/G Based Electrodes

MOFs are a sub-class of coordination polymers with poor electrical conductivity possessing a 2D or 3D porous structure that shows battery type or pseudocapacitive behavior during electrochemical analysis [10]. G-based materials are suitable for charge storage mechanisms while they possess poor specific capacitance [16].

MOF/G based composites are particularly promising electrodes in Sc application due to the possibility to achieve synergic effects between the porous solid with controlled porosity, selectivity, catalytic activity, structural tailorability, large SSA, controllable porosity, provide better ion interaction, adsorption, and desorption at the electrode–electrolyte interface, etc. and G provides conductivity, mechanical stability, high SSA, high structural stability, etc. [10,16]. The electrochemical performance of different MOF/G-based composite materials is depicted in Table 1. 

The part electrochemistry of the MOF/G based electrode included several steps in the processes of ion transport and electron transfer into/from the solid lattice [22]. The hybrid-like structure was revealed to possess the following advantages: (i) the incorporation of G, GO, and/or rGO to prevent the aggregation of the two materials; (ii) the maintenance of high electrical conductivity over the electrode; (iii) improvement of the cycling stability during the charging/discharging processes; (iv) preservation of the structural integrity of the pseudocapacitive material to prevent it from detaching from the G surface during the charge/discharge process, thus resulting in good rate capability and long cycle life [22].

A detailed discussion regarding MOF/G-based electrodes and their application towards Sc application are provided hereafter. The addition of G-based materials in MOFs has improved several properties compared to the bare materials, and this has been particularly demonstrated for MOF/G composites in Sc applications.

### 4.1. Cu-Based MOF/G Electrode

Cu^2+^ was used as a metal center in the first synthesized HKUST-1 nanomaterial. Srimuk et al. [29] performed an Sc application at the 10 wt.% rGO/HKUST-1 nanomaterial electrode. The optimized electrode showed a high BET surface area, a specific pore volume, and an average pore diameter of 1241 m^2^/g, 0.78 cm^3^/g, and 8.2 nm, respectively. The small pore size allows for easy ion transfer from the electrode-electrolyte interface of the nanomaterial. A half-cell electrode of 10 wt.% rGO/HKUST-1 coated on flexible carbon fiber paper (CFP) exhibits a high Cs of 385 F/g at 1 A/g, while pure HKUST-1 stores only 0.5 F/g. This capacitance of agglomerated rGO Sc electrode was about five-fold lower than the composite electrode. The solid-state device with a specific power of 3100 W/kg and specific energy of 42 Wh/kg supplied electricity to run a 3 V motor over a 9 min discharging time. The size of the solid-state device was about 1.8 × 4.0.

The direct preparation of highly functionalized nanosized MOFs-derived electrodes is required for assembling active electrode materials with higher SSA for electrochemical testing. He et al. [19] prepared novel copper metal MOFs-derived copper oxide@-mesoporous carbon (mC) embedded with PANI and rGO composite (CuOx/mC/PANI/rGO) by varying the pyrolysis temperature of Cu based MOFs. They reported that the ternary MOFs composite obtained at 700 °C showed the maximum Cs of 534.5 F/g at a fixed current density of 1 A/g. The improvement in Cs of the ternary composite compared to the other binary composite was ascribed to the positive synergistic effect of the ternary composite and the typically interconnected microstructure with higher electrical conductive. Initially, PANI nanorods fully and uniformly coat the porous-like frameworks of CuOx/mC nanomaterials to enhance the interface between CuOx/mC and rGO nanomaterials and increase pseudocapacitance performance. The well-interconnected rGO facilitated the higher electrolyte diffusion and exhibited better double-layer capacitance. 

Azadfalah et al. [11] prepared Cu-based MOF electrode materials composites with G (Cu-MOFs/G) for Sc application. They mentioned here that nanoflower-like nanomaterials show better electrical properties. Comparing these three electrodes to pristine Cu-MOF, Cu-MOF/G composite electrodes show the betters storage capacity, which may be due to the faradic and battery-like performances of the electrode. They reported that the values of Cs of pristine G, Cu-MOF/G composite, and Cu-MOF are 148, 482, and 190 F/g, respectively. They reported the higher values of the Cu-MOFs/G composite because porous surfaces provided better electrical properties. In addition, they reported the Cs of 65 F/g at a constant scan rate of 5 mV/s and the better cycling stability of the two-electrode device. 

### 4.2. Ni-Based MOF/G Electrode

Azadfalah et al. [37] reported a chemical synthesis of Ni-based MOF with G (Ni-MOF/G) using an in situ method, and they concluded the positive strategy effect between Ni-MOF and G materials. They reported that the values of Cs for Ni-MOF, Ni-MOF/G, and G are 660, 1017, and 148 F/g, at a scan rate of 10 mV/s, respectively. They reported the Ni-MOF/G composite offering higher vales than the other two electrodes because of the positive strategy effect between the Ni-MOF and G nanomaterial. He and his co-workers mentioned the highest cycling stability and Cs of the Ni-MOF/G nanocomposite and the possible relation between Ni-based MOF composite with G. For example, (i) the typical porous nanostructure of Ni-MOF supplied an easy method for electron transformation and provided high specific area, (ii) G materials provided a highly porous nanosheet-like nanostructure and higher electrical conductivity compared to the current collector, and (iii) the uniform-grown nanostructure and porous electrodes were more capable regarding the improvement of electrical properties and the positive synergistic effect for the nanocomposites. 

However, they reported an asymmetric device at a potential window at 1.5 V for practical purposes. The asymmetric device of Ni-MOF/G//AC shows the values of obtained Cs of 126.2 F/g with an energy density of 39.43 Wh/kg at a power density of 34.29 W/kg, respectively. In addition, they reported excellent long cycle life up to 93.5% after 1000 cycles, at 0.3 A/g current density. The improvements in electrochemical performance of the electrode were due to the higher electrical conductivity and better ion movement [37]. 

Zhou et al. [38] synthesized Ni-based MOF electrode materials for Sc using a facile chemical solvothermal method. They synthetically changed experimental parameters such as reaction temperature from 120 to 200 °C [38]. The experimental results clearly show that reaction temperatures are affected by surface morphologies, porosity, and electrical properties on the Ni-MOF electrodes. They reported that the pseudocapacitance performance of the Ni-MOFs with GO electrode material prepared at 180 °C shows better electrochemical performance. The value of Cs of the as-prepared Ni-MOF/GO composite electrode was 1457 F/g at a current density1 A/g. To improvise the electrical performance, they applied a simple strategy to the modulator Ni-MOF/GO electrode with HCl at a constant reaction temperature of 180 °C, and they obtained the highest Cs of 2192.4 F/g at the same current density (1 A/g). However, they mentioned the modulator Ni-MOF/GO composite electrode showed better cycling retention up to 85% at a constant current density of 10 A/g [38].

Kim et al. [23] synthesized Ni-based MOFs nanocomposites with the rGO by the hydrothermal method for the Sc application. They used the prepared cubic structure of the Ni-MOF/rGO electrode as a working material for Sc testing, and they reported higher values of the composite than the other electrodes. The value of the Cs of Ni-MOF/rGO electrode was 1154.4 F/g at a constant current density of 1 A/g in the 6 M KOH with better retention up to 90% at 3000 cycles [23].

Jiao et al. [17] synthesized Ni-based MOFs composite with the rGO electrode materials for the Sc application using a solvothermal method. They report the synthesis of the Ni- and Ni(OH)_2_ based MOFs composite with rGO using the solvothermal method. They obtained values of specific capacity for ADC-C/rGO, and ADC-OH/rGO electrodes were 505 and 608 C/g at a current density of 1 A/g. This considerably increased the capacity values, which may be due to the fact that the as-prepared ADC-OH/rGO electrode supplies higher electrical conductivity and the porous nanostructure of the GO nanosheet-like surface is useful for enhancing the supercapacitive performance as compared to the ADC-C/rGO. 

To use practical level application of as-prepared electrodes used as a positive electrode, and an ADC-C/rGO electrode used as a negative electrode material for HSc cell was obtained at a potential window of 0–1.5 V for various scan rates. The author reported the outstanding cycling performance attained up to 95% for 10,000 cycles at 20 A/g. This result demonstrates that the excellent behavior of ADC-OH/rGO electrode material may be attributed to the better capability rate of the positive and negative electrodes material [12]. In addition, he and his co-works have explained some features which contribute to the improvement of the electrical properties in this report.

### 4.3. Ni-Zn Based MOF/rGO Electrode

Banerjee et al. [26] reported flake-like Ni-doped MOFs (Ni-MOFs) material and composite with rGO nanosheets. They mixed a different weight percentage of the rGO (5, 10, 30, and 50%) with MOF. They claimed that the (Ni-MOFs) material uniformed distribution on all rGO nanosheets included higher concentrations of rGO material (Figure 7). Moreover, they determined the values of charge transfer resistance (R_ct_) as 184, 6, and 74 mΩ for MOFs, Ni-MOFs/rGO, and Ni-MOFs composites, respectively. This indicates the Ni-MOF-rGO materials show higher electrochemical properties than the other two electrode materials [26]. The electrochemical properties of as-prepared MOFs, Ni-MOFs, and Ni-MOFs/rGO electrode materials were tested in KOH electrolytes. The values of the Cs of the best Ni-MOFs/rGO show 758 F/gm, which is larger than the 100 F/gm (For MOFs) and 240 F/gm (rGO and Ni-MOFs) [26].

### 4.4. Co-Based MOF/GO Electrode

Hosseinian et al. [28] reported a simple way to prepare a novel composite of ZIF-67 and rGO mixed composite using a chemical route under an ultrasonic waves medium in the ethanol:water equal ratio at a room temperature. The physicochemical properties of the ZIF-67/rGO composite show better electrochemical properties as compared to the pristine ZIF-67 electrode material. The ZIF-67/rGO electrode presented higher Cs (210 F/g) than the pristine ZIF-67 electrode material (210 F/g) electrode at a 1 A/g current density, which may be due to the lower values of solution resistance (R_s_) (0.62 Ω) and R_ct_ the ZIF-67/rGO electrode compared to the pristine ZIF-67. Finally, they concluded that the Cs of ZIF-67-rGO show better electrical properties compared to the pristine ZIF-67 because of the nanoparticle-like nanostructure grown on the GO nanosheets. These types of hybrid nanostructures are more suitable for improving the electrochemical performance of the electrode [28].

### 4.5. Mo-Based MOF/G Electrode

Cao et al. [33] prepared rGO coated MoO_3_ composite with MOFs for solid-state flexible Sc using a simple and cost-effective method. After preparation of the MoO_3_ composite with rGO electrodes, prepared rGO/MoO_3_ composite materials used the positive electrode for assembly of all types of Sc devices reported in this paper. They reported the Cs of the rGO/MoO_3_ composite material is 404 F/g at a current density of 0.5 A/g. MoO_3_ electrodes prepared by directly annealing the Mo-MOFs showed poor electrochemical properties. Furthermore, they fabricated all types of flexible Sc devices for testing their electrochemical performance (Figure 8). The rGO/MoO_3_ coated MOFs electrodes offer higher Cs of flexible solid-state Sc devices with superior energy density and power density [33]. The prepared hierarchical porous nanostructures provided a high active surface area and reduced the diffusion path of ions in the electrolyte solution. In addition, the existence of rGO around metal oxide maintains a homogenous structure and provided electrically conductive networks for electron transport. Therefore, this composite owned high stability and performance [33]. 

### 4.6. Ni-Co-Based MOF/G Electrode

Rahmanifar et al. [7] synthesized Ni-MOF, Co-MOF, and Ni/Co-MOF electrodes and composited them with rGO material in the same experimental parameters as a one-pot method. They tested CV curves of Co-MOF, Co-MOF/rGO of Co-based electrode material and Ni-MOF, Ni-Co based MOF, and Ni-Co-MOF composite with rGO electrodes, respectively. The Ni/Co-MOF/rGO nanocomposite electrode material shows outstanding electrochemical performance with a Cs of 860 F/g at the 1.0 A/g. For demonstration, they fabricated an asymmetric Sc device with as-prepared Ni/Co-MOF/rGO as a positive electrode and activated carbon as a negative electrode. They observed that the asymmetric Sc device offers higher specific energy of 72.8 Wh/kg at a power density of 850 W/kg, with long cycling stability up to 91.6% after 6000 at a constant current density of 1 A/g [7]. They mentioned that the composite electrode material shows higher electrochemical performance due to various reasons, for example, the smaller size of nanostructures, the uniform growth formation on the MOFs particles, and a positive synergistic effect between the Ni and Co metal and MOFs electrode. These types of surfaces morphology provided a higher SSA of the composite material. 

Beka et al. [24] reported 2D nanosheet-like nanostructure nickel-cobalt-based MOFs (Ni-Co-MOFs) composites with rGO hybrid electrode materials for Sc by simple precipitation at room temperature. They observed that this type of hybrid surface morphology of Ni-Co-based MOFs composite with rGO electrode materials supply more SSA for faster ions transport during the electrochemical testing as well as reduce resistance and improve the current collector electrode. They reported that different affected concentrations of rGO (0, 5, 10, and 20 mg) on the electrochemical properties of the Ni-Co-MOFs electrode. The optimized Ni-Co based MOFs/rGO-2 prepared at a 10 mg concentration of the rGO composite with Ni-Co based MOFs shows the outstanding electrochemical performance with higher Cs of 1553 F/g at a lower current density of 1 A/g, may be due to the positive synergistic effect between the Ni-Co based MOFs and the different concentration of rGO electrode material. The Ni-Co-based MOFs electrode shows the Cs of 901 F/g at a 1 A/g current density of the electrode. The optimized Ni-Co-MOFs/rGO composite electrode showed better stability with 83.6% after 5000 cycles. After testing electrochemical properties, they fabricated asymmetric devices which showed a superb energy density of 44 Wh/kg at a power density of 3168 W/kg [24].

### 4.7. Fe-Based MOF/G Electrode

Graphene aerogel (GA) is superior compared to G, possessing a conductive 3D network, low density, high compressibility, and large surface area. Liu et al. [39] found that the oriented MOFs/GA composite, due to the high capacitive volume, had a faster charge/discharge rate and super cycling rate as compared with randomly distributed hexagonal prisms such as MIL-88. The SEM and TEM results of the GA and composites are shown in Figure 9. The SEM results clearly show that the GO nanosheets are uniformly coating the surface of the nanomaterial. The higher supercapacitive performance was correlated to the oriented growth and strong bonding of MOF with G. They determined Cs of the 353 F/g at the constant scan rate of 20 mV/s, with the cycling stability being 74.4% after 10,000 cycles.

### 4.8. Zn-Based MOFs/GO Electrode

An optimum quantity of mesoporous helps electrolyte penetration and easy ion transport, which is useful in facilitating high-rate electrochemical performance. The existence of micro and mesoporous is due to the evaporation of Zn nanoparticles during carbonization [32]. Zn-based MOFs (ZIF-8) and composites of G and N-doped porous carbon show outstanding electrical properties with a Cs of 245 F/g at a current density of 1 A/g in 6 M KOH electrolyte [32]. They reported two-electrode devices with a higher energy density of 18.0 Wh/kg at a power density of 475 W/kg in the wide potential window up to 1.8 V. The device reported outstanding cycle stability up to 90% capacitance retention after 10,000 cycles for this composite electrode.

Wang et al. [31] synthesized a simple and scalable highly porous carbon cubic block composite with MOFs/G for Sc application. In addition, they reported that the prepared electrode material shows an ultrafast charging rate of up to 2 V/s. They prepared three different electrodes of material for Sc such as rGO, C-MOFs, C-GMOFs. Out of these, C-GMOFs show better electrochemical results because of the lower values of resistance, which indicates higher electrical conductivity. Furthermore, they mentioned the mesoporous carbon material coated on G nanosheets, which provided a high surface area with higher active sites for electrochemical properties and an easy way to transport the ions. The optimized C-GMOFs electrode material presented the higher Sc performance with Cs of 345 F/g at 2 mV/s and outstanding retention capacitance of 99% after 10,000 cycles [31].

## 5. Conclusions and Outlook

In this review, we summarized current progress in MOF/G-based electrode materials for electrochemical Sc applications. Different characteristics, including the chemical synthesis method, transition metal-based MOF, various composites such as MOF/G, MOF/GO, MOF/rGO, and its SCs properties, have been discussed in detail. The applications of MOFs are limited due to the poor conductivity of organic linkers. To overcome conductivity issues, G-based materials are coupled with MOFs, which results in the enhancement of electrochemical characteristics. The surface engineering, TMO-MOF, three-electrode, and two-electrode electrochemical properties of the MOF/G based composites have been briefly discussed. The choice of suitable metal cations and linkers helps to tune the final structural properties for better migration of ions. Several linkers help to create covalent interaction between MOF and GO. In situ and solvothermal synthesis routes are mostly followed to ensure stronger interaction between two materials rather than the post-synthesis route. Finally, based on reported results, G-based nanostructures have shown their characteristic properties in MOF/G composites for electrochemical Sc applications.

The main advantage of MOF/G composites is the reduced diffusion pathway of migrating ions in the electrolytes, which ultimately enhances the specific capacity and rate capability of these systems. Furthermore, the restacking of G sheets can be avoided by the incorporation of MOFs between adjacent layers of G. Many researchers carry out the preparation of MOF electrodes with composite GO nanomaterials via a hydrothermal method for Sc application and obtain results that show the electrochemical performance better than the other preparation methods of MOF electrodes. However, after the hydrothermal method, researchers focused on the simple mechanical mixing chemical method for the preparation of the working MOF electrode materials. The resulting electrochemical properties decrease due to the aggregation of the G, GO, and rGO. Moreover, the direct growth of carbon-based nanomaterials such as carbon nanotubes (CNTs) and their composites with MOF electrode nanomaterials are also promising electrode materials for developing the high performance of Sc. The main challenge and issue are to improve a solid interaction between MOFs and GO instead of the weak Van der Waals force between these two nanomaterials. The major problem of GO nanosheets is their restacking, which makes a large portion of the surfaces of GO nanosheets inaccessible for charge storage and electrolyte diffusion and limits their applications. Future studies should focus on the development of flexible and mechanically stable electrodes for compact and flexible energy storage device application. 

## Figures and Tables

**Figure 1 polymers-14-00511-f001:**
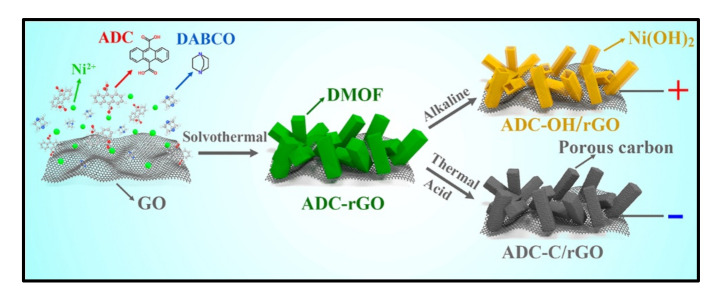
Synthesis procedures for ADC-rGO, ADC-OH/rGO, and ADC-C/rGO [17]. Reprinted with permission from Yang Jiao, Chong Qu, Bote Zhao, et al. High-Performance Electrodes for a Hybrid Supercapacitor Derived from a Metal-Organic Framework/Graphene Composite. *ACS Appl. Energy Mater.* 2019, 2, 5029–5038. Copyright (2019) American Chemical Society.

**Figure 2 polymers-14-00511-f002:**
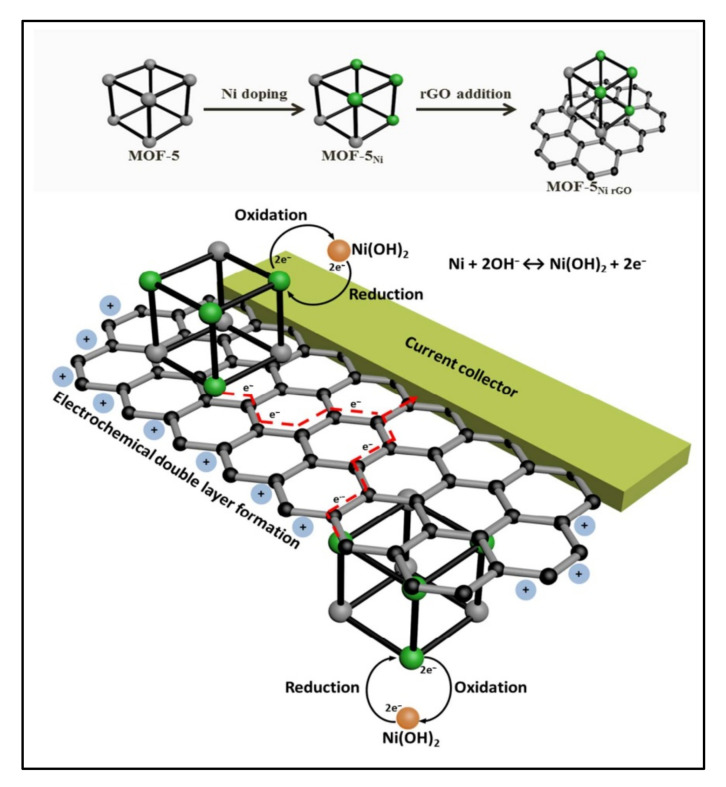
The concept for Ni doping and incorporation of rGO to form a composite of rGO and Ni-doped MOFs-5 and its use as an electrode material that harnesses both the electrochemical double layer performance of the rGO as well as the reversible redox reactions of the Ni metal centers with a synergy that facilitates effective and efficient charge transfer processes [26]. Reprinted (adapted) with permission from Parama Chakraborty Banerjee, Derrek E. Lobo, Rick Middag, et al. Electrochemical Capacitance of Ni-doped Metal-Organic Framework and Reduced Graphene Oxide Composites: More than the Sum of Its Parts. *ACS Appl. Mater. Interfaces* 2015, 7, 6, 3655–3664. Copyright (2015) American Chemical Society.

**Figure 3 polymers-14-00511-f003:**
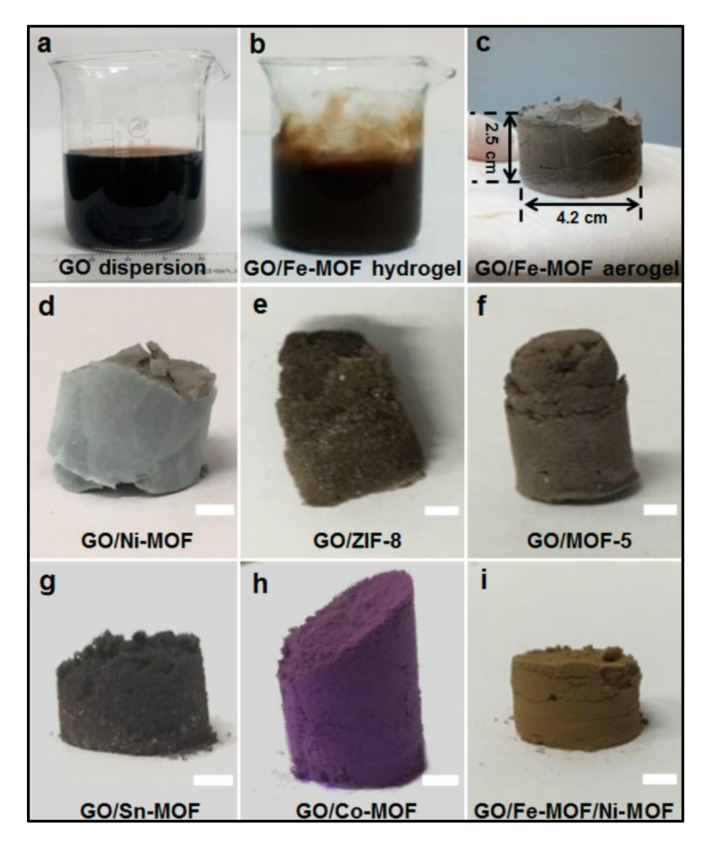
Photographs of GO dispersion (**a**), GO/Fe-MOF hydrogel (**b**), GO/Fe-MOF aerogel (**c**), and other GO/MOF composite aerogels (**d**–**i**, scale bar = 0.35 cm) [31]. Reprinted (adapted) with permission from Xilian Xu, Wenhui Shi, Peng Li, et al. Facile Fabrication of Three-Dimensional Graphene and Metal-Organic Framework Composites and Their Derivatives for Flexible All-Solid-State Supercapacitors. *Chem. Mater.* 2017, 29, 14, 6058–6065. Copyright (2017) American Chemical Society.

**Figure 4 polymers-14-00511-f004:**
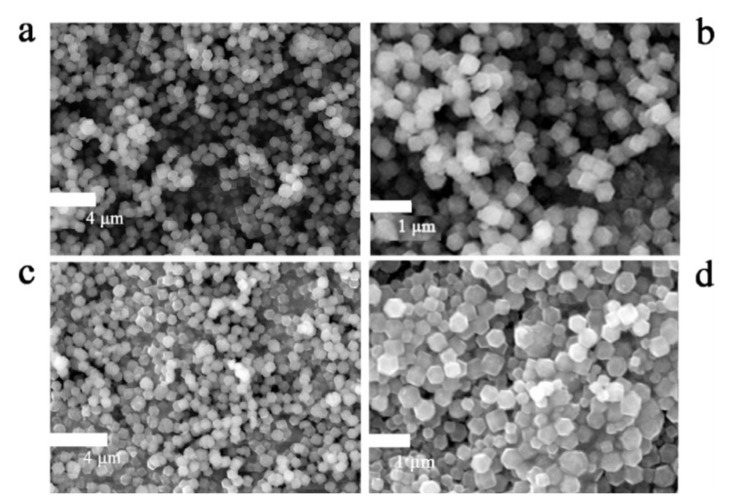
SEM (**a**–**d**) images of ZIF-8, HPNCs-800, ZIF-8/GO, and HPNCs/rGO-800. Reprinted with permission from Xin, Lijun, et al. Hierarchical metal-organic framework derived nitrogen-doped porous carbon/graphene composite for high performance supercapacitors. *Electrochim. Acta* 2017, 248, 215–224. [19] Copyrights © 2017 Elsevier Ltd. [32].

**Figure 5 polymers-14-00511-f005:**
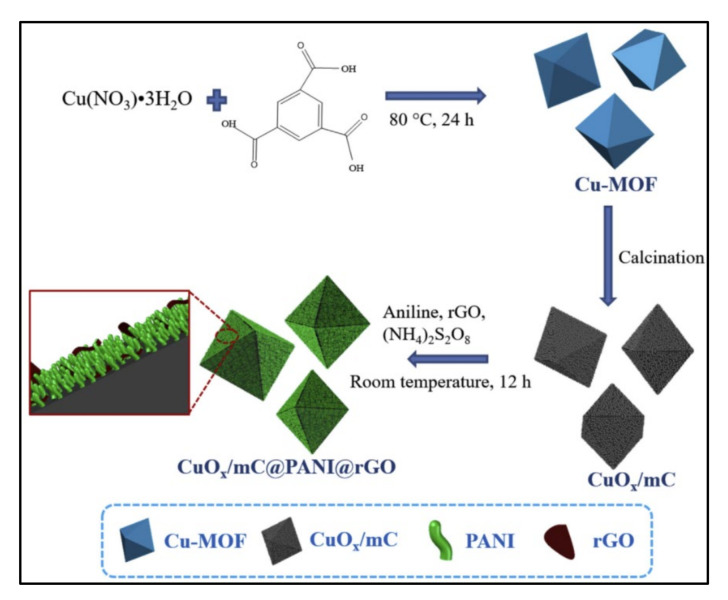
Schematic diagram of the preparation. Reprinted with permission from ref. [19] Copyrights © 2018 Elsevier Ltd.

**Figure 6 polymers-14-00511-f006:**
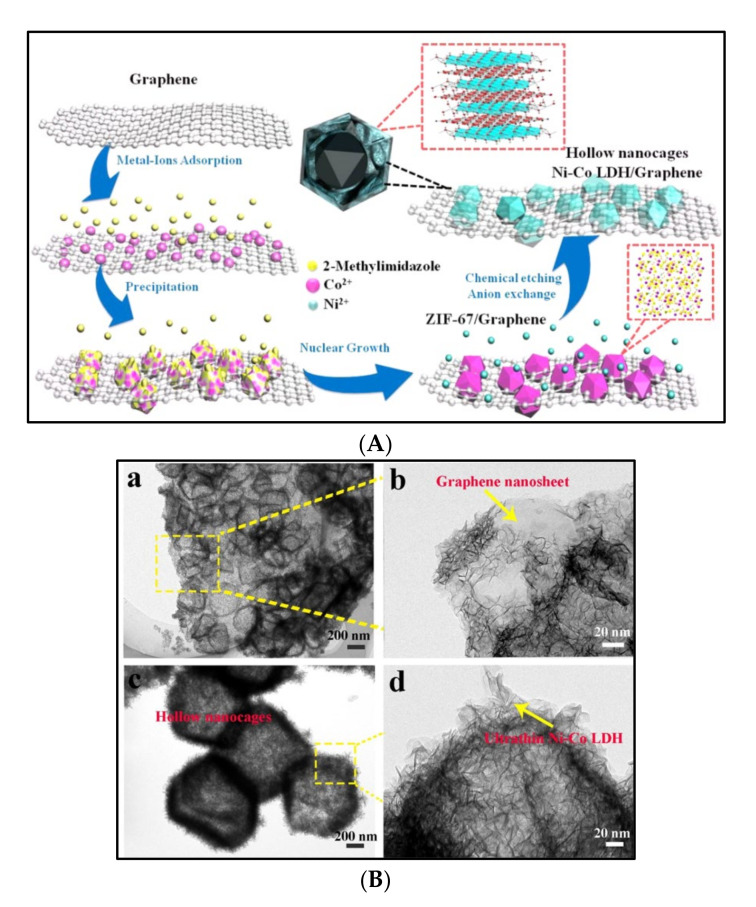
(**A**) Schematic illustration of the synthesis procedure of the Ni-Co LDH hollow nanocages/graphene composite [34]. (**B**) TEM images of Ni-Co LDH hollow nanocages/graphene composites with different magnification (**a**–**d**) [34]. Reprinted (adapted) with permission from Xue Bai, Qi Liu, Zetong Lu, et al. Rational Design of Sandwiched Ni-Co Layered Double Hydroxides Hollow Nanocages/Graphene Derived from Metal-Organic Framework for Sustainable Energy Storage. *ACS Sustainable*
*Chem. Eng.* 2017, 5, 11, 9923–9934. Copyright (2017) American Chemical Society.

**Figure 7 polymers-14-00511-f007:**
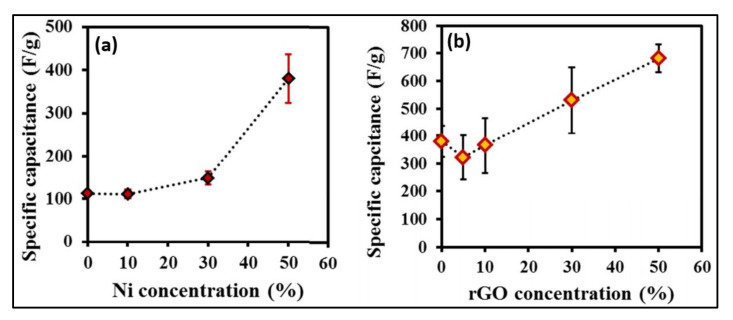
(**a**) Specific capacitance as a function of the Ni doping concentration and (**b**) specific capacitance as a function of rGO concentration in MOF-5 in 1 M KOH [26]. Reprinted (adapted) with permission from Parama Chakraborty Banerjee, Derrek E. Lobo, Rick Middag, et al. Electrochemical Capacitance of Ni-doped Metal-Organic Framework and Reduced Graphene Oxide Composites: More than the Sum of Its Parts. *ACS Appl. Mater. Interfaces* 2015, 7, 6, 3655–3664). Copyright (2015) American Chemical Society.

**Figure 8 polymers-14-00511-f008:**
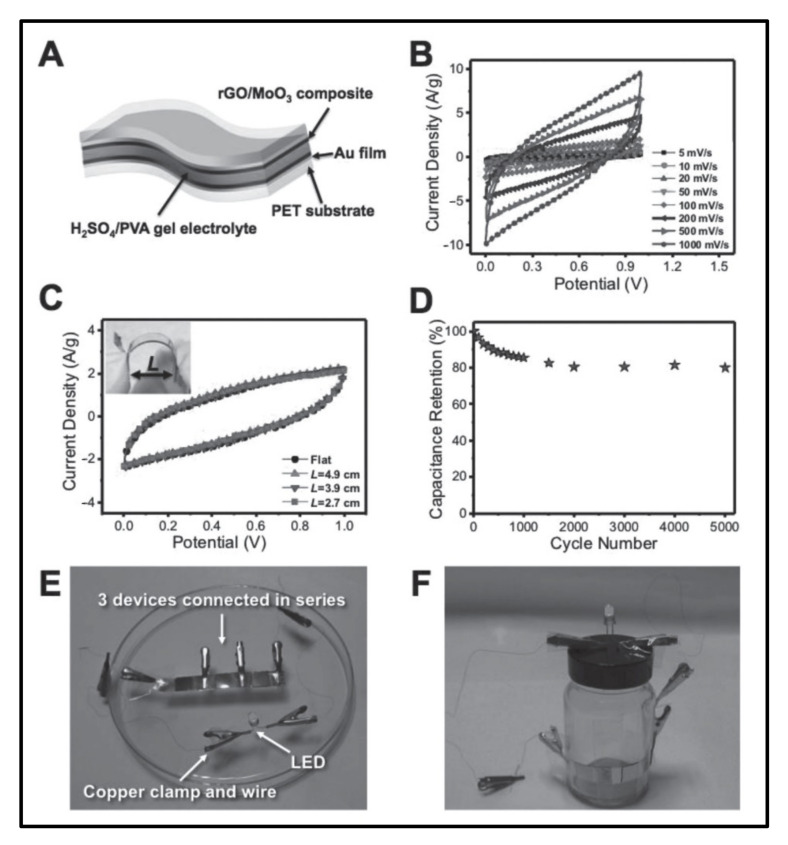
Electrochemical performance pf the all-solid-state flexible supercapacitor device fabricated with rGO/MoO_2_ composite, (**A**) Schematic illustration of the configuration of fabricated device, (**B**) CV curves of the device at different scan rates, (**C**) CV curves of the device under different bending status, (**D**) cycling performance measured by charging and discharging the device at 2 A/g for 5000 cycles, (**E**,**F**) real photographs of the device [33]. Reprinted with permission from Cao, Xiehong, et al. Reduced graphene oxide-wrapped MoO_3_ composites prepared by using metal-organic frameworks as precursor for all-solid-state flexible supercapacitors, *Adv. Mater.* 2015, 27, 32, 4695–4701. Copyright © 2015 WILEY-VCH Verlag GmbH & Co. KGaA, Weinheim [33].

**Figure 9 polymers-14-00511-f009:**
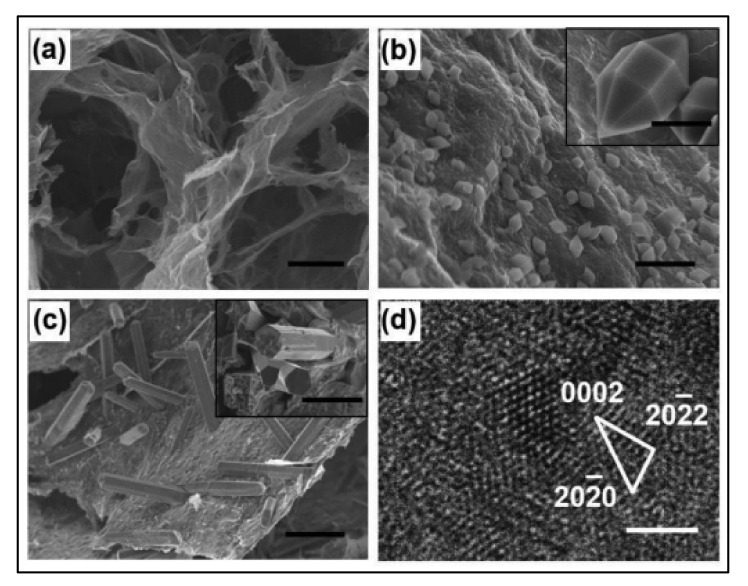
(**a**–**c**) SEM images of the GA, GAMOF, GAMOF composite, (**d**) TEM of the GAMOF. Reprinted with permission from Liu, Lin, et al. Growth-oriented Fe-based MOFs synergized with graphene aerogels for high-performance supercapacitors. *Adv. Mater. Interfaces* 2018, 5, 8, 1701548. Copyright © 2018 WILEY-VCH Verlag GmbH & Co. KGaA, Weinheim [39].

**Table 1 polymers-14-00511-t001:** Comparison of electrochemical performances for MOF/G-based Sc.

Electrode Material	Electrode/MOF Details	Energy Density (Wh/kg)	Power Density (W/kg)	Cycling Stability (%)/Cycles	Method	Electrolyte	Specific Capacitance (F/g)	Ref.
C-POMOF@rGO	Polyoxometallate-MOF	20.10	9071	94, 5000	Hydrothermal	6 M KOH	178 (0.5 A/g)	[22]
rGO/Fe_2_O_3_	MOF-derived composite aerogels	79.2	405	96.3, 5000	Mixing	6 M KOH	869.2 (1 A/g)	[30]
GO/Mo-MOF	Mo-MOF, imidazole	55	400	87.5, 6000	Mixing	1 M H_2_SO_4_	617 (1 A/g)	[33]
NiCo-MOF/rGOja	2-methylimidazole	44	3168	83.6, 5000	Precipitation	-	1553 (1 A/g)	[24]
Ni-BPDC/CNF	Biphenyl dicarboxylic acid and carbon nanofiber	48.1	1064.7	92, 5000	Precipitation	4 M KOH	250.6 mA h/g (1 A/g)	[35]
C@GQD	ZIF-8, G quantum dots	-	-	90, 10,000	Precipitation	5 M KOH	137 (1 A/g)	[36]
HPNCs/rGO-800	ZIF-8, G, and N-doped porous carbon	18.0	475	90, 10,000	Precipitation	6 M KOH	245 (1 A/g)	[32]
CuO_x_@mC@PANI@rGO	(MOF)-derived copper oxide@mesoporous carbon (CuO_x_@mC)	-	-	70, 500	Polymerization	1 M H_2_SO_4_	534.5 (1 A/g)	[19]
C-GMOF	MOF-5/GO derived carbon, BDC	30.30	137	99, 10,000	Reflux	6 M KOH	345 (2 mV/s)	[31]
Ni/Co-MOF-rGO	TMU-10, 4,4′-Oxybis-benzoic acid	72.8	42.5	91.6, 6000,	Reflux	6 M KOH	860 (1 A/g)	[7]
Ni-Co LDH ZIF-67	Ni-Co layered double hydroxides on G nanosheets	68	594.9	92.9, 2000	Reflux	1 M KOH	1265 (1 A/g)	[34]
ADC-C/rGO	Ni_2_(ADC)_2_(DABCO)/rGO 9,10-anthracenedicarboxylate and 1,4-diazabicyclo octane	59	872	95, 10,000	Solvothermal	2 M KOH	330 (1 A/g)	[17]
Cu-MOF/G	Benzene-1,3,5-tricarboxylic acid (BTC)	34.5	1350	93.8, 1000	Solvothermal	6 M KOH	482 (10 mV/s)	[11]
Ni-MOF/rGO	Trimesic acid (H_3_BTC)	0.82	20	90, 3000	Solvothermal	6 M KOH	1154.4 (1 A/g)	[23]
Ni-MOF/GO	1,4-dicarboxybenzene (H_2_BDC)	39.43	34.29	93.5, 1000	Solvothermal	6 M KOH	1017 (10 mV/s)	[37]
Ni-MOFs@GO		-	-	85.1, 3000	Solvothermal	2 M KOH	2192.4 (1 A/g)	[38]
rGO/HKUST-1	HKUST-1	3100	42	98.5, 4000	Ultrasonication	0.5 M Na_2_SO_4_	85 (1 A/g)	[29]
rGO-Ni-MOF	Ni-doped MOF, BDC	37.8	227	59, 500	Ultrasonication	1 M KOH	758 (1.28 F/cm^2^)	[26]
rGO/ZIF-67	2-methylimidazole	-	-	80, 1000	Ultrasonication	6 M KOH	210 (1 A/g)	[28]

## Data Availability

All the data used in this study is already included in the manuscript.

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
