# Peer review of "MOFs-Graphene Composites Synthesis and Application for Electrochemical Supercapacitor: A Review"

_polymers, 2022, doi:10.3390/polym14030511_

Round 1

Reviewer 1 Report

The review covers the investigations related to the synthesis of the MOF – graphene composites and to their application in the assembly of electrochemical supercapacitors. The authors tried to adequately present variety of the methods used and classified the achievements on the base of the MOF structure and their combination with the metals and graphene related materials. Although the topic of the review is very interesting and deserves the publication, the quality of the manuscript seems to be rather low and should be significantly improved.

  1. First and foremost, English quality does not meet standards of scientific publications.
  2. Second, the authors should present more information about the structure and classification of the MOFs including specific acronyms like ZIF etc.
  3. Other reviews in the related area should be provided and the necessity in another one should be substantiated in the Introduction
  4. The authors should definitely show the relations between the specific structure of the modifier and the characteristics of supercapacitors.
  5. The titles of the sections are sometimes confusing (3.1 Solvothermal – what? 3.6 In situ – what?, 4. Electrochemical properties of MOF-Graphene electrodes – capacitors are not the same as electrodes)
  6. Many figures illustrating the mechanism of action or the synthesis of MOF structures are either trivial or not informative. It is suggested to reconsider their necessity (figs. 4, 7, 8, 9c)
  7. Please check the use of acronyms – many of them are not explained, some are exhausting and can be omitted.

Author Response

Date: 2022-01-18

Response to the Reviewers Comments

Dear Editor,

Greetings:

Sincerest thanks for your response and reviewer’s comments on our manuscript. Following your letter regarding the manuscript “MOFs-Graphene Composites Synthesis and Application for Electrochemical Supercapacitor: A Review” revised submission to Polymers for publication. We found the comments very helpful and constructive. We have modified the paper in response to the extensive and insightful reviewer comments and the main changes were marked using the track change’s function. We have rewritten a few sections of the manuscript and we hope that this complies with the referee’s remarks.

Reviewer #1

The review covers the investigations related to the synthesis of the MOF – graphene composites and to their application in the assembly of electrochemical supercapacitors. The authors tried to adequately present variety of the methods used and classified the achievements on the base of the MOF structure and their combination with the metals and graphene related materials. Although the topic of the review is very interesting and deserves the publication, the quality of the manuscript seems to be rather low and should be significantly improved.

The authors are very thankful to the reviewer for appreciating and approving our work. We have addressed all the comments and revised the manuscript thoroughly. The suggested changes have been incorporated into the revised manuscript and the relevant changes are highlighted.

  1. First and foremost, English quality does not meet standards of scientific publications.

Response: Thank you for your comments. We acknowledge the mistake noted by the reviewer. We have revised the manuscript carefully to omit grammatical errors.

  1. Second, the authors should present more information about the structure and classification of the MOFs including specific acronyms like ZIF etc.

Response: Thank you for your comment. We have corrected it as you pointed out.

  1. Other reviews in the related area should be provided and the necessity in another one should be substantiated in the Introduction

Response: Thank you for your comments. We have included relevant review articles in the introduction section. To the best of our knowledge, there are very few reviews on MOF-graphene-based electrochemical supercapacitors. However, some related reviews have been included in the introduction section.

  1. The authors should definitely show the relations between the specific structure of the modifier and the characteristics of supercapacitors.

Response: Thank you for your comments. We have corrected it as you pointed out.

  1. The titles of the sections are sometimes confusing (3.1 Solvothermal – what? 3.6 In situ – what?, 4. Electrochemical properties of MOF-Graphene electrodes – capacitors are not the same as electrodes)

Response: Thank you for your comments. We acknowledge the mistake noted by the reviewer. We have corrected it as you pointed out.

  1. Many figures illustrating the mechanism of action or the synthesis of MOF structures are either trivial or not informative. It is suggested to reconsider their necessity (figs. 4, 7, 8, 9c)

Response: Thank you for your comments. We have removed Fig. 4, 8, and 9c in the revised manuscript.

  1. Please check the use of acronyms – many of them are not explained, some are exhausting and can be omitted.

Response: Thank you for your comments. We have corrected it as you pointed out.

Reviewer 2 Report

This work reports about recent advancement in MOFs-GO composites synthesis and their application as supercapacitor.

Hereafter a few comments:

  • The introduction does not focus on the materials and does not provide the state-of-the-art concerning MOFs, GO, and supercapacitors;
  • graphene and GO (graphene oxide?) are used as if they refer to the same material, which is not accurate;
  • the synthesis section does not highlight the strengths and weaknesses of the different techniques. In several cases, the starting material is not clearly stated, eg GO structure, as well as the composite structures;
  • the electrochemical properties section reports mostly the capacitance and power densities, while several other properties are neglected. Such properties are not linked to the device different structures and materials, metal properties/GO nanomaterials properties, and its nanostructure;
  • the importance of MOFs-GO stated in the conclusion is clear, but the authors fail to support the advantages offered by such an approach compared to GO nanomaterials themself

A thorough language check is necessary to improve the writing quality.

Author Response

Date: 2022-01-18

Response to the Reviewers Comments

Dear Editor,

Greetings:

Sincerest thanks for your response and reviewer’s comments on our manuscript. Following your letter regarding the manuscript “MOFs-Graphene Composites Synthesis and Application for Electrochemical Supercapacitor: A Review” revised submission to Polymers for publication. We found the comments very helpful and constructive. We have modified the paper in response to the extensive and insightful reviewer comments and the main changes were marked using the track change’s function. We have rewritten a few sections of the manuscript and we hope that this complies with the referee’s remarks.

Reviewer #2

This work reports about recent advancement in MOFs-GO composites synthesis and their application as supercapacitor.

Hereafter a few comments:

The authors are very thankful to the reviewer for appreciating and approving our work. We have addressed all the comments and revised the manuscript thoroughly. The suggested changes have been incorporated into the revised manuscript and the relevant changes are highlighted.

  1. The introduction does not focus on the materials and does not provide the state-of-the-art concerning MOFs, GO, and supercapacitors;

Response: Thank you for your comments and suggestions. We have revised the introduction section as per your comments. Several review articles based on MOFs and MOFs-derived metal oxides for supercapacitor application have been published in the past several years. However, there are very few reviews on MOF-graphene-based electrochemical supercapacitors. However, some related reviews have been included in the introduction section to provide the present state of the art.

  1. Graphene and GO (graphene oxide?) are used as if they refer to the same material, which is not accurate;

Response: Thank you for your comments. We acknowledge the mistake noted by the reviewer. We have corrected it as you pointed out.

  1. The synthesis section does not highlight the strengths and weaknesses of the different techniques. In several cases, the starting material is not clearly stated, eg GO structure, as well as the composite structures;

Response: The authors are very thankful to the reviewer for the comments and the authors are agree with the reviewers and hence to improve the quality of our review the advantages and weaknesses of the chemical synthesis methods have been included in the revised manuscript.

  1. The electrochemical properties section reports mostly the capacitance and power densities, while several other properties are neglected. Such properties are not linked to the device different structures and materials, metal properties/GO nanomaterials properties, and its nanostructure.

Response: Thank you for your comment and the authors are agree with the reviewer’s opinion and comment. After serious revision of manuscript, literature survey, and considering reviews comments we noticed that the very few papers, review paper available in the literature. This review is based on the novel MOF, graphene (G), reduced graphene oxide (rGO), graphene oxide (GO), and their composite for electrochemical application. In the revised manuscript, we have focused to cover all properties of electrode material.

  1. The importance of MOFs-GO stated in the conclusion is clear, but the authors fail to support the advantages offered by such an approach compared to GO nanomaterials themselves.

Response: Thank you for you comments. We have revised conclusion section and included advantages of MOF/G systems in the revised manuscript.

  1. A thorough language check is necessary to improve the writing quality.

Response: Thank you for your comments. We have corrected as you pointed out.

Reviewer 3 Report

  1. The abstract is too short to highlight the purpose of the article, I suggest the author condense the meaning of the review.
  2. I think the review needs catalogs.
  3. The introduction lacks logic and the content is too single. I think it should be contained following aspects: (1) Detailed background research. (2) EDLC and pseudocapacitance. (3) Metal organic frameworks (MOF). (4) Synthetic methods for MOFs-graphene.
  4. You need to enrich your introduction with figures.
  5. The introduction should describe the specific work in detail.
  6. The section of “Electrochemical capacitor” should be involved in the introduction.
  7. The part of synthesis methods just lists the research work without further detailed analysis, you must point out the advantages and disadvantages and analyze the future development trend.
  8. The part of “Electrochemical properties of MOF-Graphene electrodes” only introduces the mixed metal MOF in supercapacitors. I think the role of MOF composites in supercapacitors regarding specific capacitance, energy and Power density should be added.
  9. The conclusion is unsubstantiated.
  10. There are too few references to meet the review criteria.
  11. The language of the review needs polishing up.

Author Response

Date: 2022-01-18

Response to the Reviewers Comments

Dear Editor,

Greetings:

Sincerest thanks for your response and reviewer’s comments on our manuscript. Following your letter regarding the manuscript “MOFs-Graphene Composites Synthesis and Application for Electrochemical Supercapacitor: A Review” revised submission to Polymers for publication. We found the comments very helpful and constructive. We have modified the paper in response to the extensive and insightful reviewer comments and the main changes were marked using the track change’s function. We have rewritten a few sections of the manuscript and we hope that this complies with the referee’s remarks.

Reviewer #3

The authors are very thankful to the reviewer for appreciating and approving our work. We have addressed all the comments and revised the manuscript thoroughly. The suggested changes have been incorporated into the revised manuscript and the relevant changes are highlighted.

  1. The abstract is too short to highlight the purpose of the article, I suggest the author condense the meaning of the review.

Response: Thank you for the reviewer’s comment. As per the reviewer’s suggestion, the abstract section has been thoroughly revised in order to improve the quality of the manuscript, and its detailed explanation is included in the abstract section.

  1. I think the review needs catalogs.

Response: Thank you for your comments and suggestions. We have included table of contents and list of abbreviations in the revised manuscript.

  1. The introduction lacks logic and the content is too single. I think it should be contained the following aspects: (1) Detailed background research. (2) EDLC and pseudocapacitance. (3) Metal-organic frameworks (MOF). (4) Synthetic methods for MOFs-graphene.

Response: Thank you for your comments and suggestions. We have revised the introduction section as per your suggestions.

  1. You need to enrich your introduction with figures.

Response: Thank you for the reviewer’s comment. We are agreed with your comments and suggestions. However, another reviewer recommended removing some figures so in the present mini-review we cannot include more figures in the introduction. We assure you that we will consider your suggestions for our future work. We have included sufficient figures in the revised manuscript and rewritten the introduction section.

  1. The introduction should describe the specific work in detail.

Response: Thank you for the reviewer’s comment. According to the reviewer’s suggestions, more details, clarity, and scientific perspective are included in the text.

  1. The section of “Electrochemical capacitor” should be involved in the introduction.

Response: Thank you for the reviewer’s comment. We are agreed with our comments however the structure of the review will be disturbed if we merge the “Electrochemical capacitors” section in the introduction section. However, we have revised our introduction and described EDLC and pseudocapacitors in short.

  1. The part of synthesis methods just lists the research work without further detailed analysis, you must point out the advantages and disadvantages and analyze the future development trend.

Response: Thank you for the reviewer’s comment and the authors are agree with the reviewer’s opinion and suggestions. As per the reviewer’s suggestion, the part of synthesis methods has been thoroughly revised. The advantages and limitations of synthesis methods are included in the revised manuscript.  

  1. The part of “Electrochemical properties of MOF-Graphene electrodes” only introduces the mixed metal MOF in supercapacitors. I think the role of MOF composites in supercapacitors regarding specific capacitance, energy, and Power density should be added.

Response: Thank you for your valuable suggestion, as per the reviewer’s suggestion, we have corrected the revised manuscript.

  1. The conclusion is unsubstantiated.

Response: Thank you for your comment. We agree to the reviewer’s comment. As per the reviewer’s suggestion, we have rewritten the conclusion in the revised manuscript, and its detailed explanation is included in the conclusion section.

  1. 10. There are too few references to meet the review criteria.

Response: Thank you for the reviewer’s comment. As per the reviewer’s suggestion, the detailed progression has been thoroughly revised in order to improve the quality of the manuscript, but there are very few reports are published on the MOF/G based composites for electrochemical supercapacitors.

  1. 11. The language of the review needs polishing up.

Response: Thank you for the reviewer’s comment. The manuscript was further corrected for grammar and made necessary corrections were in order to improve the quality of the manuscript. We have also re-written some sentences for clarifying the contents.

Round 2

Reviewer 1 Report

I agree with the changes made and believe the manuscript can be submitted for publication in present form; the only minor comment - please check the subscripts in the chemical formulae in the titles of the refs. [2] and [6]. 

Author Response

Date: 2022-01-21

Response to the Reviewers Comments

Dear Editor,

Greetings:

Sincerest thanks for your response and reviewer’s comments on our manuscript. Following your letter regarding the manuscript “MOFs-Graphene Composites Synthesis and Application for Electrochemical Supercapacitor: A Review” submitted to Polymers for publication. We found the comments very helpful and constructive. We have modified the paper in response to the extensive and insightful reviewer comments and the main changes were marked using the track change’s function. We have rewritten a few sections of the manuscript and we hope that this complies with the referee’s remarks.

Reviewer #1

I agree with the changes made and believe the manuscript can be submitted for publication in present form; the only minor comment - please check the subscripts in the chemical formulae in the titles of the refs. [2] and [6].

Response: Thank you for your comment. We have corrected it as you pointed out.

Reviewer 2 Report

I thank the authors for this new version which takes into account most of the previous comments.

Section 2 and Fig 1 look a bit out of scope, I would suggest removing them since the knowledge of SC is taken for granted in Section 1.

I would like them to stress more the advantages/limitations of each technique presented in Section 3 and the relation between synthesis/electrode deposition with SC performances.

English is quite improved, but far from being satisfactory.

Author Response

Date: 2022-01-21

Response to the Reviewers Comments

Dear Editor,

Greetings:

Sincerest thanks for your response and reviewer’s comments on our manuscript. Following your letter regarding the manuscript “MOFs-Graphene Composites Synthesis and Application for Electrochemical Supercapacitor: A Review” submitted to Polymers for publication. We found the comments very helpful and constructive. We have modified the paper in response to the extensive and insightful reviewer comments and the main changes were marked using the track change’s function. We have rewritten a few sections of the manuscript and we hope that this complies with the referee’s remarks.

Reviewer #2

I thank the authors for this new version which takes into account most of the previous comments.

The authors are very thankful to the reviewer for appreciating and approving our work. We have addressed all the comments and revised the manuscript thoroughly. The suggested changes have been incorporated into the revised manuscript and the relevant changes are highlighted.

  1. Section 2 and Fig 1 look a bit out of scope, I would suggest removing them since the knowledge of SC is taken for granted in Section 1.

Response: Thank you for your comments. We have deleted Figure 1 in the revised manuscript.

  1. I would like them to stress more the advantages/limitations of each technique presented in Section 3 and the relation between synthesis/electrode depositions with SC performances.

Response: Thank you for your comments and suggestions. We have included advantages and disadvantages for all synthesis methods. We have correlated the synthesis method and electrochemical supercapacitor performance of the MOF/G-based composites in the revised manuscript. The available reports suggest the direct influence of the synthetic method on electrochemical properties. The morphological features of the materials, amount of graphene, the nature of polymeric binders, linkers, and the synthesis method have also been systematically proven to be factors influencing the electrochemical supercapacitor properties of the composites.

  1. English is quite improved, but far from being satisfactory.

Response: Thank you for your comments. We have carefully revised the manuscript to avoid grammar errors.
